# Long-range DNA end resection supports homologous recombination by checkpoint activation rather than extensive homology generation

Michael T Kimble[1,2], Matthew J Johnson[1,2], Mattie R Nester[2], Lorraine S Symington[2,3]*

[1]Program in Biological Sciences, Columbia University, New York, United States; [2]Department of Microbiology & Immunology, Columbia University Irving Medical Center, New York, United States; [3]Department of Genetics & Development, Columbia University Irving Medical Center, New York, United States

**Abstract** Homologous recombination (HR), the high-fidelity mechanism for double-strand break (DSB) repair, relies on DNA end resection by nucleolytic degradation of the 5′-terminated ends. However, the role of long-range resection mediated by Exo1 and/or Sgs1-Dna2 in HR is not fully understood. Here, we show that Exo1 and Sgs1 are dispensable for recombination between closely linked repeats, but are required for interchromosomal repeat recombination in *Saccharomyces cerevisiae*. This context-specific requirement for long-range end resection is connected to its role in activating the DNA damage checkpoint. Consistent with this role, checkpoint mutants also show a defect specifically in interchromosomal recombination. Furthermore, artificial activation of the checkpoint partially restores interchromosomal recombination to *exo1Δ sgs1Δ* cells. However, cell cycle delay is insufficient to rescue the interchromosomal recombination defect of *exo1Δ sgs1Δ* cells, suggesting an additional role for the checkpoint. Given that the checkpoint is necessary for DNA damage-induced chromosome mobility, we propose that the importance of the checkpoint, and therefore long-range resection, in interchromosomal recombination is due to a need to increase chromosome mobility to facilitate pairing of distant sites. The need for long-range resection is circumvented when the DSB and its repair template are in close proximity.

**\*For correspondence:**
lss5@cumc.columbia.edu

**Competing interest:** The authors declare that no competing interests exist.

## Editor's evaluation

This study shows that long-range resection is important for recombination between distal, but not proximal, homologous sequences. It is thus proposed that a major role of long resection of a double strand break mediated by Sgs1 and Exo1 is to activate the DNA damage checkpoint to allow the chromosomal mobility needed for the DNA ends to find a distant homologous sequence for repair via homologous recombination. Consistently, *exo1 sgs1* mutants show defects specifically in inter-chromosomal homologous recombination. The study adds a new biological meaning to the role of long DNA resection, providing a new mechanism for the control of homologous recombination.

## Introduction

DNA double-stand breaks (DSBs) pose a threat to genome integrity and cell viability. If unrepaired, DSBs can lead to cell death, and if repaired improperly, DSBs can lead to loss of genetic information or chromosomal rearrangements associated with various pathologies such as neurodegeneration and

cancer (*Moynahan and Jasin, 2010*; *Rass et al., 2007*). Cells have two primary pathways to repair DSBs: homologous recombination (HR) and non-homologous end joining (NHEJ). HR is the high-fidelity mode of DSB repair because it uses a homologous template, generally the sister chromatid, for repair. HR is essential for maintaining genome integrity. For example, there is increased sensitivity to DNA damaging agents and increased frequencies of chromosome rearrangements in HR-deficient yeast cells (*Putnam and Kolodner, 2017*; *Symington et al., 2014*). Furthermore, defects in human HR proteins, such as BRCA1 and BRCA2, are associated with increased risk for breast and ovarian cancers, as well as Fanconi anemia (*Prakash et al., 2015*; *Taylor et al., 2019*; *Wang et al., 2015*).

To repair DSBs by HR, the Mre11-Rad50-Xrs2/NBS1 (MRX/N) complex, stimulated by CDK-phosphorylated Sae2/CtIP, nicks the 5′-terminated strands on either side of the DSB, followed by 3′–5′ exonucleolytic processing back towards the break ends (*Cannavo and Cejka, 2014*; *Garcia et al., 2011*). This end-clipping reaction is followed by further processing from the 5′ ends by Exo1 or Dna2-Sgs1/BLM to create longer 3′ overhangs (*Cejka and Symington, 2021*). This second step, termed long-range resection, has been shown to be required for several HR processes, including single-strand annealing (SSA), yeast mating-type switching and interchromosomal gene conversion (*Gobbini et al., 2020*; *Guo et al., 2017*; *Mimitou and Symington, 2008*; *Zhu et al., 2008*). In *Saccharomyces cerevisiae*, resection initiation by MRX is essential to process DSBs with end-blocking lesions, whereas clean DSBs can be processed directly by the long-range resection machinery in the absence of MRX, albeit with delayed kinetics (*Cejka and Symington, 2021*). Once initiated, resection proceeds at about 4 kb/h in cells that lack a homologous template for repair (*Chung et al., 2010*; *Zhu et al., 2008*). Measurements of resection tract lengths in cells undergoing recombination have primarily utilized site-specific endonucleases, which cleave both sister chromatids, thus preventing use of the preferred donor duplex to template HR. By using a donor allele that lacks the endonuclease cleavage site, either in diploids or haploids with repeats on different chromosomes, average resection tracts of >2 kb in length were reported and resection tract length correlated with the time required for repair (*Chung et al., 2010*). Consistent with this finding, a more recent study found that resection tracts were short for a rapid sister chromatid repair event (*Jakobsen et al., 2019*).

Although long-range resection is part of canonical models of repair by HR, the physiological role of this process is not completely understood given that several lines of evidence suggest minimal homology or reduced resection is sufficient for HR. For example, spontaneous recombination can occur with ~250 bp of homology (*Jinks-Robertson et al., 1993*). Additionally, efficient gene conversion can occur with as little as 250 bp of homology on either side of a programmed DSB (*Inbar et al., 2000*), which is within the range of resection tract lengths produced by MRX-catalyzed short-range resection (*Cannavo et al., 2019*; *Gnügge et al., 2023*; *Mimitou et al., 2017*). Taken together, these results indicate that recombination can occur through much shorter tracts of ssDNA than are produced by long-range resection. Indeed, it has been shown that in G2-phase *exo1Δ sgs1Δ* cells, repair efficiency in response to ionizing radiation is only slightly reduced and kinetics are delayed by around one hour compared to wild-type (WT) (*Westmoreland and Resnick, 2016*). Furthermore, diploids lacking Exo1 or its nuclease activity exhibit near WT levels of joint molecule formation and meiotic divisions even though resection tract lengths are greatly reduced (~270 nucleotides in *exo1-nd* cells compared with ~800 nucleotides in WT; *Mimitou et al., 2017*; *Zakharyevich et al., 2010*). These findings suggest that long-range resection may not be necessary for HR in all scenarios; however, the reason for this context dependence remains unclear.

In addition to generating ssDNA substrates for Rad51-catalyzed HR, the end resection machinery is associated with DNA damage checkpoint signaling (*Waterman et al., 2020*). The MRX/N complex recruits and activates the Tel1/ATM kinase in response to DSBs, after which signaling transitions to Mec1/ATR once resection generates sufficient ssDNA for RPA and Ddc2/ATRIP binding (*Shiotani and Zou, 2009*; *Waterman et al., 2020*). Once activated, the DNA damage checkpoint limits extensive resection to prevent accumulation of excessive ssDNA through multiple mechanisms. Rad9/Crb2 antagonizes the Dna2-Sgs1 mechanism in yeast (*Bonetti et al., 2015*; *Ferrari et al., 2015*; *Lazzaro et al., 2008*; *Leland et al., 2018*), while Rad53, the effector kinase for Tel1 and Mec1, inhibits Exo1 activity by phosphorylation of the C-terminal regulatory domain (*Morin et al., 2008*; *Yu et al., 2018*). The 9-1-1 DNA damage clamp (Ddc1, Mec3, and Rad17 in budding yeast) and Rad24, the large subunit of the 9-1-1 clamp loader complex, attenuate resection by MRX and promote resection by Exo1 (*Gobbini et al., 2020*; *Ngo et al., 2014*; *Ngo and Lydall, 2015*). The Tel1 and Mec1 kinases

positively influence resection initiation by MRX and Sae2 but limit extensive resection by promoting recruitment and retention of Rad9 to chromosome in the vicinity of DSBs (*Cannavo et al., 2018*; *Cartagena-Lirola et al., 2006*; *Waterman et al., 2020*).

The DNA damage checkpoint and end resection are also linked to chromosome mobility. DNA damage induces mobility of the broken chromosome (local), and, to a lesser extent, mobility of undamaged chromosomes (global; *Dion et al., 2012*; *Miné-Hattab and Rothstein, 2012*; *Seeber et al., 2013*). The checkpoint proteins Mec1 and Rad9 are necessary for local and global DNA damage-induced chromosome mobility (*Dion et al., 2012*; *Seeber et al., 2013*). Additionally, checkpoint activation in the absence of DNA damage is sufficient to increase chromosome mobility (*Seeber et al., 2013*). Chromosome mobility is also linked to end resection, as deletion of Sae2, which delays resection, also delays chromosome mobility (*Miné-Hattab and Rothstein, 2012*). Therefore, resection, checkpoint activation, and chromosome mobility likely collaborate to facilitate DNA repair, especially when a repair template is not in close proximity to the broken chromosome.

Given that the requirement for long-range resection seems to be context dependent, we wanted to explore further the role of long-range resection in HR. Here, we show that interchromosomal gene conversion is significantly impaired in the absence of long-range resection, consistent with previous findings (*Gobbini et al., 2020*; *Guo et al., 2017*). Remarkably though, we find that intrachromosomal recombination between closely linked repeated sequences occurs at almost WT levels when long-range resection is eliminated, providing evidence that long-range resection is dispensable under certain circumstances. Thus, the ssDNA tracts exposed by MRX-Sae2 alone must be sufficient to facilitate HR, and the requirement for long-range resection in the interchromosomal context must go beyond simply exposing adequate ssDNA. We suggest that the main role for long-range resection in mediating interchromosomal recombination is in DNA damage checkpoint activation. When the DSB and repair template are spatially separated, cells require time and a more active homology search facilitated by chromosome mobility, and therefore long-range resection is necessary. When the DSB and repair template are in close proximity, the homology search is rapid enough that the need for checkpoint activation and chromosome mobility is circumvented, and long-range resection is dispensable.

## Results

### Long-range resection is necessary for interchromosomal recombination but dispensable for intrachromosomal recombination

To investigate the need for long-range resection in different contexts, we employed two different reporter systems designed to measure intrachromosomal or interchromosomal HR. The intrachromosomal assay system measures recombination between two non-functional *ade2* alleles (*ade2-I* and *ade2-n*), separated by a 4.3 kb sequence containing the *TRP1* gene integrated at the native *ADE2* locus on Chr XV (*Figure 1A*; *Mozlin et al., 2008*). The *ade2-I* allele was generated by replacement of the endogenous *Aat*II site with an I-*Sce*I cut site, and *ade2-n* contains a frame shift mutation at the *Nde*I site. A cassette expressing the I-*Sce*I endonuclease under the control of a galactose-inducible promoter was integrated at the *LYS2* locus on Chr II. The assay for measuring interchromosomal recombination contains the same *ade2-I* allele at the *ADE2* locus on Chr XV and galactose-inducible I-*Sce*I cassette on Chr II, but the *ade2-n* repair template is integrated at the *LEU2* locus on Chr III (*Figure 1B*). In both assays, after induction of I-*Sce*I, cells must repair the DSB introduced at the *ade2-I* locus in order to survive, and most do so using the *ade2-n* donor allele; thus, survival frequency is a measure of recombination efficiency. The surviving colonies can be categorized as Ade$^+$/Ade$^-$ by colony color (Ade$^+$=white; Ade$^-$=red) and Trp$^+$/Trp$^-$ by growth on SC -Trp plates. The *ADE2* and *TRP1* status of surviving colonies in the intrachromosomal assay is indicative of the pathway used for repair, including gene conversion with or without crossover, break-induced replication (BIR), or SSA (*Figure 1—figure supplement 1*). For the interchromosomal assay, the two *ade2* alleles are oriented such that crossovers are viable, although we do not differentiate between crossovers and non-crossovers here. We only determine *ADE2* status in this assay.

Using these recombination assays, we found that WT cells can repair DSBs with high efficiency in both contexts. Mre11 is important for HR in both contexts, with an increased role in interchromosomal repair (*Figure 1C and D*). Interestingly, *mre11Δ* cells show an increase in Trp$^-$ recombinants in the

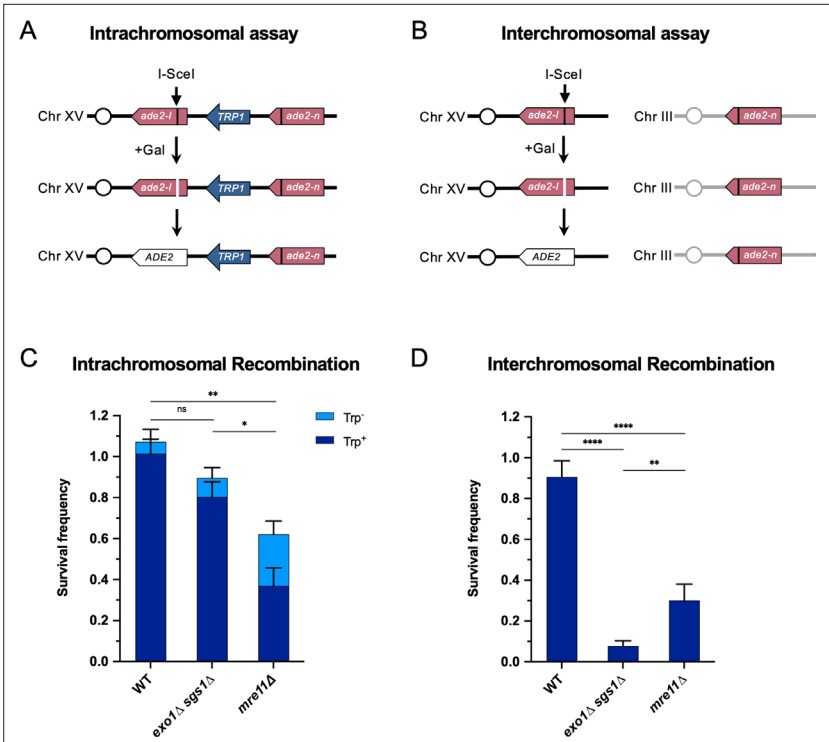

**Figure 1.** Long-range end resection is necessary for interchromosomal repair but not for intrachromosomal repair. (**A**) Representation of the intrachromosomal assay. The reporter contains an *ade2-I* allele with an I-*Sce*I recognition site and an *ade2-n* allele with a frameshift mutation oriented as direct repeats on Chr XV. The vertical black lines indicate the sites of the mutations. The galactose-inducible I-*Sce*I endonuclease is integrated at the *LYS2* locus on Chr II (not shown). After break formation, the *ade2-I* allele uses the *ade2-n* allele to restore a functional *ADE2* allele, most of the time accompanied by retention of the *TRP1* marker. Other outcomes are shown in *Figure 1—figure supplement 1*. (**B**) The interchromosomal assay contains the same *ade2* alleles as in the intrachromosomal assay, except *ade2-n* is located on Chr III. Survival frequency in response to constitutive I-*Sce*I expression for the intrachromosomal strains (**C**) and interchromosomal strains (**D**). Intrachromosomal repair products are categorized as Trp+ or Trp-. Bars represent mean values from at least 3 plating assays per genotype. Error bars represent standard deviation. Significance values are indicated by: ns- not significant, * $p<0.05$, ** $p<0.01$, *** $p<0.001$, **** $p<0.0001$ based on a two-tailed t-test. Source data are available in *Figure 1—source data 1*.

The online version of this article includes the following source data and figure supplement(s) for figure 1:

**Source data 1.** This file contains all the source data for *Figure 1* and related figure supplements.

**Figure supplement 1.** Each of the long-range resection nucleases can independently facilitate intrachromosomal repair.

**Figure supplement 2.** Each of the long-range resection nucleases can independently facilitate Rad51-independent intrachromosomal repair.

**Figure supplement 3.** Each of the long-range resection nucleases can independently facilitate interchromosomal repair.

intrachromosomal repair assay, which may be explained by Mre11's role in coordinating resection at both ends of a break and suppressing BIR (*Pham et al., 2021*; *Westmoreland and Resnick, 2013*). In the absence of Mre11, one end of the DSB may invade the *ade2-n* allele, followed by BIR synthesis toward the telomere, thereby deleting *TRP1*. Nearly all of these Trp- events are also Ade+, which fits with this prediction (*Figure 1—figure supplement 1*). Intrachromosomal recombination in the absence of Rad51, in which SSA is the primary mode of repair, also showed a moderate dependency on Mre11 (*Figure 1—figure supplement 2*). Consistent with an SSA mode of repair, the *TRP1* marker was deleted in nearly all such repair events (*Figure 1—figure supplement 2*).

The requirement for Exo1 and Sgs1 in interchromosomal repair has been shown previously (*Guo et al., 2017*; *Lydeard et al., 2010*; *Zhu et al., 2008*), and we also find that long-range resection is

required for efficient interchromosomal recombination in our assay (*Figure 1D*). However, long-range resection is largely dispensable for intrachromosomal recombination (*Figure 1C*). The *exo1Δ* or *sgs1Δ* single mutants have minimal effects on SSA, intra- or interchromosomal recombination frequency (*Figure 1—figure supplement 1*; *Figure 1—figure supplement 2*; *Figure 1—figure supplement 3*). The one notable change in the type of repair outcome is the increase in Ade⁺ Trp⁻ events for intrachromosomal recombination in the absence of Sgs1 (*Figure 1—figure supplement 1*). This increase in Trp⁻ events is likely attributable to Sgs1's role in dissolution of recombination intermediates (*Ira et al., 2003*), since crossover events in our system can lead to deletion of *TRP1*. Given that the homology shared between the DSB and repair alleles (3.7 kb) is the same in both assays, the recombination defect of *exo1Δ sgs1Δ* cells in the interchromosomal context is likely due to something other than a failure to expose sufficient homology.

## Interchromosomal repair is slower than intrachromosomal repair and is coupled to DNA damage checkpoint activation

We next determined repair kinetics and checkpoint activation in both assays. We reasoned that interchromosomal repair may be relatively slow, leading to accumulation of ssDNA and activation of the DNA damage checkpoint. If checkpoint activation is required to facilitate repair, this could explain the interchromosomal recombination defect in the checkpoint-defective *exo1Δ sgs1Δ* cells (*Gravel et al., 2008*; *Zhu et al., 2008*). A PCR-based assay was employed to measure repair product accumulation over the first 8 hr after DSB induction (*Figure 2A*). After the *ade2-I* allele is repaired and converted to *ADE2*, an *Aat*II restriction site is restored where the I-*Sce*I recognition site had been inserted. Therefore, PCR amplification of the recipient allele followed by *Aat*II digestion in vitro detects recombination events.

Using this assay, we evaluated repair kinetics of intra- and interchromosomal recombination in WT and *exo1Δ sgs1Δ* backgrounds. Intrachromosomal repair in WT cells was slightly faster than interchromosomal repair or intrachromosomal repair in *exo1Δ sgs1Δ* cells (*Figure 2B–C*). While the WT cells reached the same level of intra- and interchromosomal repair by the 6 hr timepoint, *exo1Δ sgs1Δ* intrachromosomal recombination levels lagged behind throughout the time course, only reaching levels comparable to WT at 8 hr. *exo1Δ sgs1Δ* cells did not show detectable levels of interchromosomal recombination throughout the 8 hr time course (*Figure 2B–C*), consistent with the strong reduction in survival observed in plating assays (*Figure 1D*). It should also be noted that delayed repair kinetics of *exo1Δ sgs1Δ* cells may be partially due to a defect in DSB formation, although this defect does not completely account for the difference in repair kinetics (*Figure 2—figure supplement 1*).

Rad53 phosphorylation was monitored as a readout of checkpoint activation at the same timepoints used for the recombination assays. Checkpoint activation was robust in the WT interchromosomal strain but absent in the intrachromosomal strain (*Figure 2D*). This result suggests that the relatively short delay (~0.5–1 hr) in interchromosomal repair leads to checkpoint activation. These findings are consistent with previous studies showing that Rad53 is not activated during mating-type switching, a fast, intrachromosomal DSB repair process, but is activated during interchromosomal ectopic repair or when the recombination enhancer for mating-type switching is deleted (*Kim and Haber, 2009*; *Mehta et al., 2017*; *Pellicioli et al., 2001*).

Others have shown that checkpoint activation in response to a DSB is abrogated in the absence of long-range resection (*Balogun et al., 2013*; *Bantele et al., 2019*; *Gobbini et al., 2020*; *Gravel et al., 2008*; *Zhu et al., 2008*). We also evaluated checkpoint activation during intra- and interchromosomal repair in the absence of Exo1 and Sgs1. Consistent with what other groups have reported, there was a defect in DSB-induced Rad53 phosphorylation in *exo1Δ sgs1Δ*, although there was a low level of phosphorylation even before DSB induction that persisted throughout the time course (*Figure 2D*). Any shift in Rad53 mobility after DSB induction is comparable between the intra- and interchromosomal *exo1Δ sgs1Δ* strains, which is not the case in the WT strains, indicating that the phosphorylation detected is not associated with slower repair. The DSB-independent Rad53 phosphoshift in *exo1Δ sgs1Δ* cells requires the checkpoint proteins Rad9 and Rad24 (*Figure 2—figure supplement 2*). Therefore, in the absence of Exo1 and Sgs1 there is likely ssDNA accumulation throughout the genome, possibly arising during replication, which triggers low-level checkpoint activation. Taken together, these results support the idea that interchromosomal repair is slower than intrachromosomal repair, allowing enough ssDNA to accumulate to activate the checkpoint, which then promotes repair.

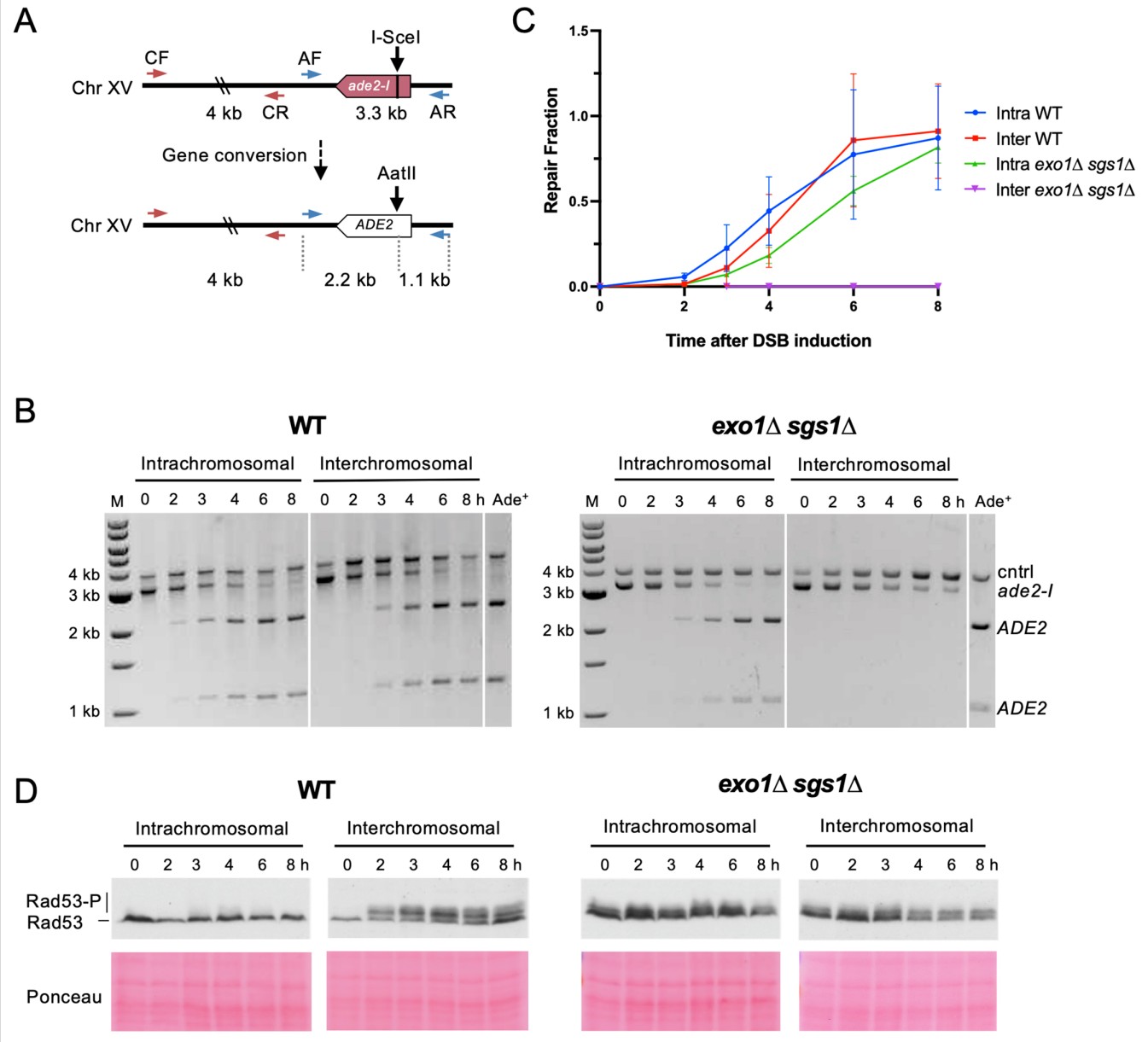

**Figure 2.** Interchromosomal repair is slower than intrachromosomal repair and is coupled to checkpoint activation. (**A**) Schematic for the PCR assay used to measure repair in both the intra- and interchromosomal strains. Primers AF (MK193) and AR (MK197) amplify the *ade2/ADE2* allele. These two primers generate a 3.3 kb product before repair, and 2.2 kb and 1.1 kb products after repair and digestion with *Aat*II. Primers CF (MK238) and CR (MK239) were used as a control and generate a 4 kb product, regardless of repair status. (**B**) Representative results for the PCR-based assay. DNA from an Ade+ colony was used as a reference for 100% repair. M refers to 1 kb size ladder (New England BioLabs). Time after DSB induction is indicated. (**C**) Quantification of the repair products for the assay shown in B. Mean of three biological replicates is plotted and error bars represent standard deviation. (**D**) Western blots to detect Rad53 phosphorylation (top) and corresponding Ponceau S staining (bottom). Source data are available in *Figure 2—source data 1*.

The online version of this article includes the following source data and figure supplement(s) for figure 2:

**Source data 1.** This file contains all the source data for *Figure 2* and related figure supplements.

**Figure supplement 1.** I-SceI cutting kinetics.

**Figure supplement 2.** Rad53 phosphorylation in the absence of Exo1 and Sgs1 is dependent on Rad9 and Rad24.

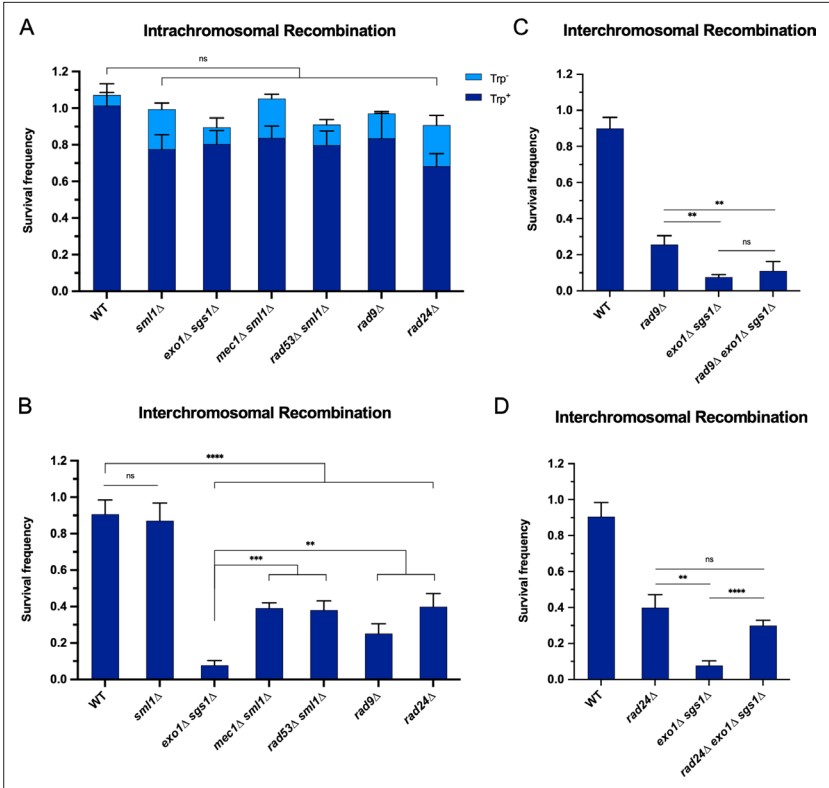

**Figure 3.** The requirement for long-range resection in interchromosomal recombination correlates with an increased requirement for the DNA damage checkpoint. Survival frequency in the plating assay for the intrachromosomal strains (**A**) and interchromosomal strains (**B–D**) with the indicated genotypes. Intrachromosomal repair products are categorized as Trp$^+$ or Trp$^-$. Bars represent mean values from at least three plating assays per genotype. Error bars represent standard deviation. Significance values are indicated by: ns- not significant, ** p<0.01, *** p<0.001, **** p<0.0001 based on a two-tailed t-test. (**C and D**) contain overlapping data with B, so only relevant statistics are shown. Source data are available in *Figure 3—source data 1*.

The online version of this article includes the following source data and figure supplement(s) for figure 3:

**Source data 1.** This file contains all the source data for *Figure 3* and related figure supplements.

**Figure supplement 1.** Loss of *RAD24*, but not *RAD9* suppresses the resection defect of *exo1Δ sgs1Δ* cells.

However, in the absence of long-range resection, the checkpoint is not properly activated during interchromosomal repair, potentially leading to progression through the cell cycle before repair of the DSB.

## The requirement for long-range resection in interchromosomal recombination correlates with an increased requirement for the DNA damage checkpoint

We hypothesized that if the checkpoint defect of *exo1Δ sgs1Δ* contributes to the interchromosomal recombination defect, then checkpoint mutants should also exhibit a defect in interchromosomal recombination, while minimally influencing intrachromosomal recombination. Indeed, we found that *mec1Δ sml1Δ*, *rad53Δ sml1Δ*, *rad9Δ*, and *rad24Δ* mutants were proficient for intrachromosomal HR but exhibited a significant decrease in interchromosomal recombination (*Figure 3A and B*). It is worth noting that the decrease in interchromosomal recombination in these checkpoint mutants is independent of any long-range resection deficiency since resection in the *rad9Δ* and *mec1Δ sml1Δ* mutants is more efficient than WT (*Bonetti et al., 2015*; *Clerici et al., 2014*; *Lazzaro et al., 2008*), and the *rad24Δ* mutant exhibits mild or no resection defect (*Aylon and Kupiec, 2003*; *Gobbini et al., 2020*). Furthermore, *rad9Δ* is epistatic to *exo1Δ sgs1Δ* (*Figure 3C*), indicating that the checkpoint defect due to loss of Rad9 has no further effect on recombination

efficiency. As mentioned above, the Rad53 phosphorylation observed in *exo1Δ sgs1Δ* cells prior to DSB induction was Rad9 dependent (*Figure 2—figure supplement 2*). This suggests that prolonged checkpoint activation in *exo1Δ sgs1Δ* cells is likely not the reason for the interchromosomal recombination defect since the reduction in Rad53 activation in *exo1Δ sgs1Δ rad9Δ* cells does not result in increased recombination efficiency (*Figure 3C*), consistent with a recent report (*Gobbini et al., 2020*).

It is also noteworthy that the checkpoint mutants had a milder interchromosomal repair defect than the *exo1Δ sgs1Δ* cells (*Figure 3B*). This difference may be accounted for by the combined effects of checkpoint loss and shorter resection tracts in *exo1Δ sgs1Δ* cells. In support of this idea, combining *rad24Δ* with *exo1Δ sgs1Δ*, which has recently been shown to increase Mre11-dependent resection (*Gobbini et al., 2020*), partially rescued the interchromosomal recombination deficiency of *exo1Δ sgs1Δ* cells to near *rad24Δ* levels (*Figure 3D*). We also confirmed that Rad24 loss increases resection in *exo1Δ sgs1Δ* cells up to ~1.3 kb from a DSB using a qPCR-based resection assay (*Figure 3—figure supplement 1*; *Zierhut and Diffley, 2008*). However, *rad9Δ* did not rescue the resection defect of *exo1Δ sgs1Δ* cells, which is consistent with the failure to rescue the recombination defect (*Figure 3C*, *Figure 3—figure supplement 1*). Therefore, the *exo1Δ sgs1Δ* interchromosomal recombination defect is primarily due to a checkpoint defect, with partial contribution from critically short resection tracts.

## Restoration of the checkpoint restores interchromosomal recombination efficiency in exo1 sgs1 cells

We next asked whether restoration of the DNA damage checkpoint could rescue interchromosomal recombination defects of checkpoint-deficient cells. It has previously been shown that expression of a Ddc2-Rad53 fusion protein suppresses the checkpoint defect and DNA damage sensitivity of *rad9Δ* cells (*Lee et al., 2004*). Indeed, we found that expression of Ddc2-Rad53 was able to restore interchromosomal recombination efficiency of *rad9Δ* cells to near WT levels (*Figure 4A*). Thus, restoration of the DNA damage checkpoint can lead to a corresponding rescue of recombination efficiency.

However, we found no rescue, and even a slight decline, of interchromosomal recombination upon expression of the Ddc2-Rad53 fusion in *exo1Δ sgs1Δ* cells (*Figure 4—figure supplement 1*), presumably because the resection tracts are too short to support recruitment of Ddc2-Rad53. Notably, the fusion did not affect resection tract length, indicating that the lower recombination in *exo1Δ sgs1Δ* cells with Ddc2-Rad53 versus without Ddc2-Rad53 is not due to impaired resection (*Figure 4—figure supplement 1*). DSB formation as measured by HO cutting in the resection assay is slower in the *exo1Δ sgs1Δ* Ddc2-Rad53 cells, but this should not lead to lower survival in the recombination assay (*Figure 4—figure supplement 1*). We reasoned that extending resection tracts by deleting *RAD24* (*Gobbini et al., 2020*) might lead to a more efficient rescue by the Ddc2-Rad53 fusion. Although the fusion was unable to improve recombination in the *exo1Δ sgs1Δ rad24Δ* mutant (*Figure 4—figure supplement 1*), the *rad24Δ* mutation resulted in a slight suppression of the *exo1Δ sgs1Δ* Ddc2-Rad53 recombination defect. The Ddc2-Rad53 fusion did not rescue the *rad24Δ* interchromosomal recombination defect, likely because of the failure to robustly activate Mec1 in the absence of Mec1 activators Ddc1 and Dpb11 at the DSB site without loading of the 9-1-1 complex (*Bonilla et al., 2008*; *Majka et al., 2006*; *Mordes et al., 2008*; *Navadgi-Patil and Burgers, 2008*).

As an alternative strategy, we employed a Ddc1-Ddc2 co-localization system that has previously been shown to activate the checkpoint, even in the absence of DNA damage (*Bonilla et al., 2008*). Briefly, upon addition of galactose to the medium, cells express Ddc1-LacI and Ddc2-LacI fusions that are recruited to a LacO array, thereby co-localizing the 9-1-1 complex (Ddc1-Mec3-Rad17) and Mec1 (via Ddc2) to artificially activate the checkpoint. Importantly, checkpoint activation in this system does not rely on ssDNA generation. Using this system and measuring interchromosomal repair by the PCR assay, we found that the HR defect of the *exo1Δ sgs1Δ* strain was partially rescued and Rad53 phosphorylation was modestly restored (*Figure 4B and C*). One technical limitation of the artificial checkpoint system is that the Ddc1- and Ddc2-LacI fusions are over-expressed from *GAL* promoters, and induction of longer than ~2 hr has a dominant negative effect on Rad53 activation (D. Toczyski, personal communication). Since I-*Sce*I is also expressed from the *GAL1-10* promoter, reduced I-*Sce*I cutting resulting from the shorter induction time could contribute to lower HR efficiency. Interestingly, a longer galactose induction (5 hr) resulted in delayed repair in the WT strain, consistent with the

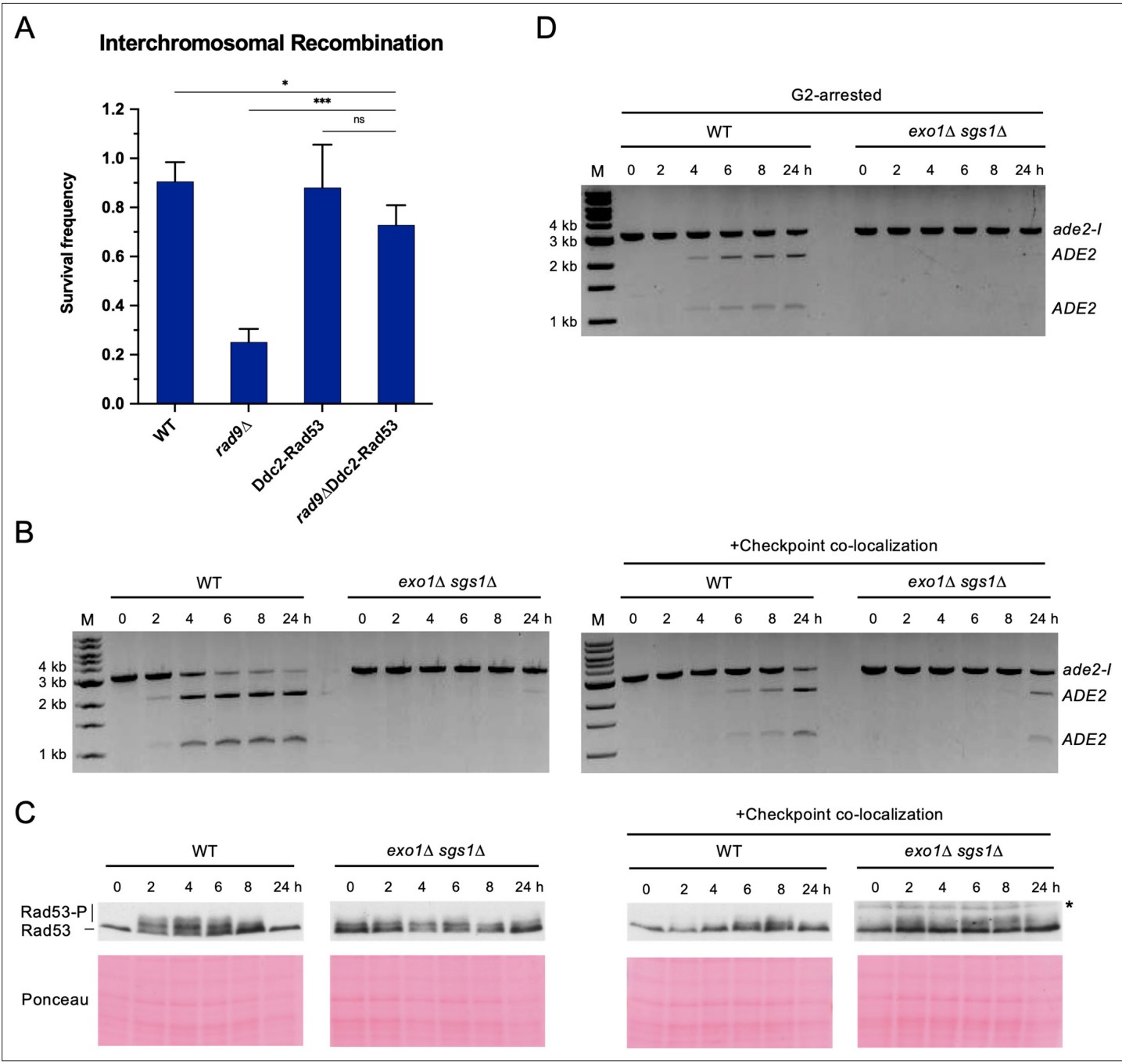

**Figure 4.** Restoring the checkpoint can rescue an interchromosomal recombination defect. (**A**) Survival frequency in the plating assay for the interchromosomal strains with the indicated genotypes. Bars represent mean values from at least three plating assays per genotype. Error bars represent standard deviation. Significance values are indicated by: ns- not significant, * p<0.05, *** p<0.001, based on a two-tailed t-test. (**B**) Representative results from the PCR assay in interchromosomal assay strains with and without the Ddc1/Ddc2 co-localization system. The DSB and checkpoint were transiently induced with galactose for 3 hr and cells were collected at the indicated time points. (**C**) Western blots for Rad53 phosphorylation (top) and corresponding Ponceau S staining (bottom) for strains with and without the Ddc1/Ddc2 co-localization system. The asterisk denotes a non-specific band detected by the HA antibody. (**D**) Representative result from the PCR assay in G2-arrested cells. WT and *exo1Δ sgs1Δ* strains containing the interchromosomal assay were arrested with nocodazole for 2 hr prior to transient (3 hr) DSB induction (t$_o$). Samples were collected at the indicated time points. For (**B**) and (**D**), M refers to 1 kb size ladder (New England BioLabs). Source data are available in *Figure 4—source data 1*.

The online version of this article includes the following source data and figure supplement(s) for figure 4:

**Source data 1.** This file contains all the source data for *Figure 4* and related figure supplements.

**Figure supplement 1.** Checkpoint restoration in *exo1Δ sgs1Δ* interchromosomal assay strains.

**Figure supplement 2.** Artificial checkpoint restoration in *exo1Δ sgs1Δ* interchromosomal strains.

dominant negative effect on Rad53 phosphorylation and the need for checkpoint activation for repair (*Figure 4—figure supplement 2*).

One function of the DNA damage checkpoint is to halt cell cycle progression in the presence of unrepaired damage. Therefore, we attempted to extend the G2/M phase of *exo1Δ sgs1Δ* cells by addition of nocodazole to the growth medium prior to I-*Sce*I induction, but this failed to alleviate the interchromosomal recombination defect (*Figure 4D*). Notably, nocodazole treatment was unable to rescue *rad9Δ* and *rad24Δ* interchromosomal recombination defects in two previous studies (*Aylon and Kupiec, 2003*; *Ferrari et al., 2020*). Thus, the interchromosomal recombination defect of checkpoint-deficient cells cannot be attributed solely to cell division prior to completion of repair. It is likely that the importance of the checkpoint is due to a role in promoting some other process as well.

## Repair template proximity affects ectopic repair efficiency

The DNA damage checkpoint has been shown to regulate chromosome mobility in response to DSBs and this function has been suggested to enhance recombination between distant recombining sites (*Dion et al., 2012*; *Miné-Hattab and Rothstein, 2012*). Thus, we considered the possibility that the homology search using an interchromosomal donor is more spatially challenging than using an intrachromosomal donor and is more dependent on chromosome mobility. Analysis of previously published Hi-C data (*Duan et al., 2010*; *Lazar-Stefanita et al., 2017*) showed that total contacts between the DSB allele and the interchromosomal site on Chr III ranked in the bottom 10% among all *ADE2* contacts genome-wide. We sought to alleviate this spatial challenge by bringing the DSB and repair alleles in closer proximity, which has previously been shown to improve recombination in WT cells (*Agmon et al., 2013*; *Lee et al., 2016*). We chose two interchromosomal sites with higher contact frequencies (on Chr IV and XVI) and an additional site with a low contact frequency (on Chr VIII). Recombination assays were carried out with these strains in both WT and *exo1Δ sgs1Δ* backgrounds. In all cases, there was a significant defect in the *exo1Δ sgs1Δ* strains compared to the respective WT strains. There was a weak trend towards improved interchromosomal recombination in the absence of long-range resection by using a higher contact donor (*Figure 5—figure supplement 1*). However, the recombination efficiency at these sites was still substantially lower than intrachromosomal recombination, likely due to the fact that even what we qualified as high contact frequency interchromosomal sites are relatively low contact compared to intrachromosomal sites.

We next explored whether moving the donor allele further away from the recipient allele on the same chromosome influenced the dependence on long-range resection for recombination. The intrachromosomal strains that have been used in all prior experiments have the *ade2-n* donor located ~4 kb from the *ade2-I* allele. We integrated *ade2-n* into six additional locations across Chr XV and measured recombination efficiency in both WT and *exo1Δ sgs1Δ* backgrounds. Again, survival frequency remained high for all donor locations in the WT strains (*Figure 5A*). However, survival frequency showed a proximity-based effect in the *exo1Δ sgs1Δ* strains. The two sites located within 20 kb of the broken allele facilitated efficient repair, independent of long-range resection (*Figure 5A*). The donor located 54 kb from the DSB showed a mild dependency on long-range resection, and beyond 100 kb on either chromosome arm, cells were more dependent on long-range resection for recombination (*Figure 5A*). Spearman correlation analysis confirmed that survival frequency showed a negative correlation with the linear distance between donor and recipient ($r=-0.857$; $p=0.024$), and a positive correlation with total contacts according to the Lazar-Stefanita Hi-C data ($r=0.821$; $p=0.034$; *Lazar-Stefanita et al., 2017*). Both comparisons fit to one-phase decay models (*Figure 5B*).

When examining Rad53 phosphorylation in several of these new intrachromosomal strains, we found that the checkpoint was not activated in a strain where the donor was located 19 kb from the DSB. However, we detected weak Rad53 phosphorylation in the strain where the donor was located 54 kb from the DSB, and more robust Rad53 phosphorylation for the 448 kb donor (*Figure 5C*). Therefore, checkpoint activation correlated with the requirement for long-range resection, as was observed for interchromosomal recombination. These findings demonstrate that long-range resection-independent recombination occurs over a relatively short distance (~50 kb) within the same chromosome. Beyond this distance, checkpoint activation, and therefore long-range resection, are necessary to promote efficient recombination.

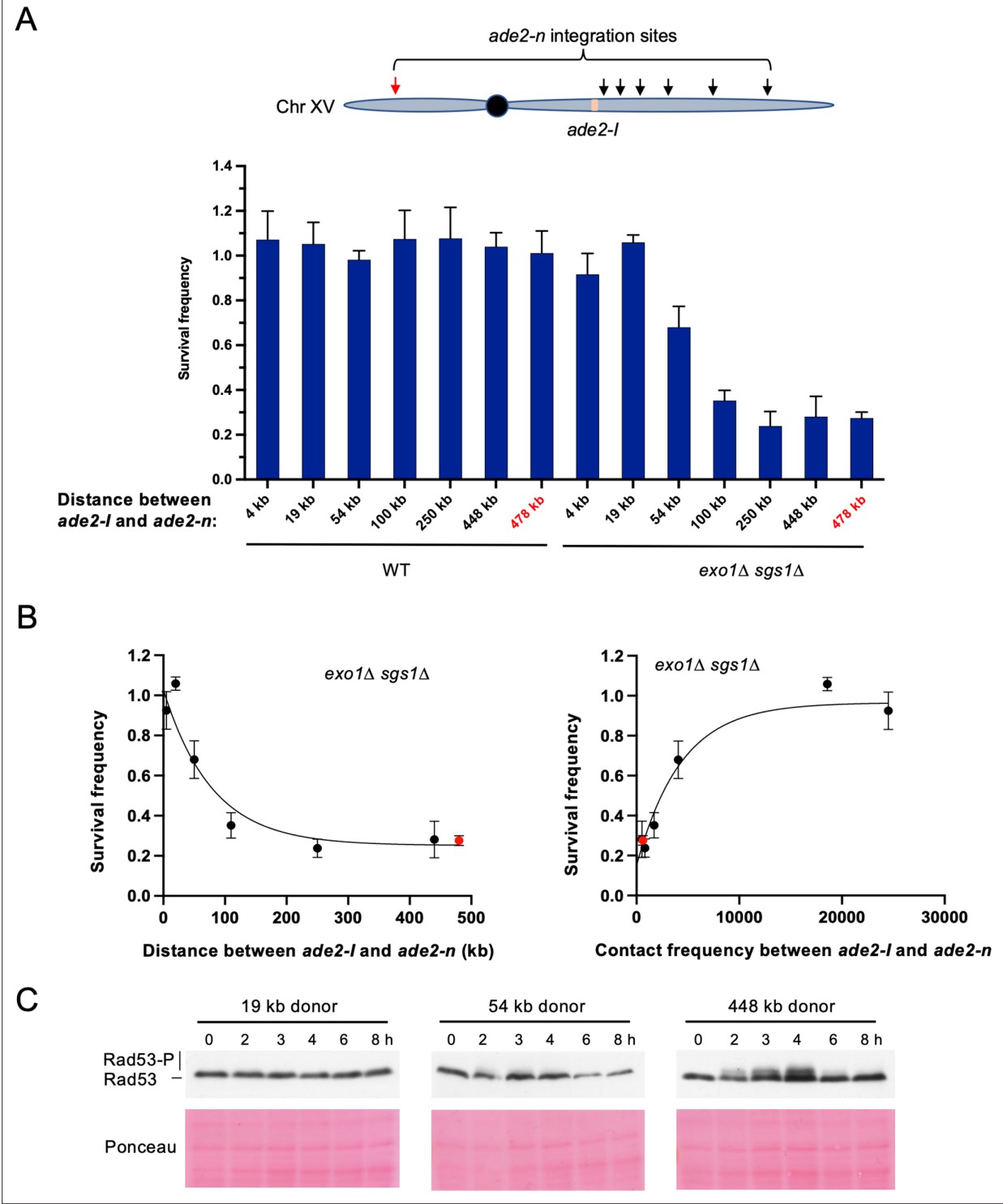

**Figure 5.** Repair template proximity dictates the requirement for long-range resection. (**A**) Schematic of Chr XV with the *ade2-n* insertion sites indicated with arrows and the *ade2-I* DSB allele in beige (top). Below are the survival frequencies for the plating assay for intrachromosomal strains. On the left are survival frequencies for WT strains and on the right are survival frequencies for *exo1Δ sgs1Δ* strains. The donor site on the left arm of Chr XV is indicated in red. The 4 kb donor is the intrachromosomal site used in all other experiments. Bars represent mean values from at least three plating assays per genotype. Error bars represent standard deviation. (**B**) Correlations between survival and linear distance between *ade2-I* and *ade2-n* alleles (left) and between survival and total contact frequency between *ade2-I* and *ade2-n* (right) (*exo1Δ sgs1Δ* strains). Nonlinear, one-phase decay regressions were applied in Prism, which are represented by the curves in each graph. Total contact frequencies are based on data from *Lazar-Stefanita et al., 2017*. (**C**)

*Figure 5 continued on next page*

*Figure 5 continued*

Western blots for Rad53 phosphorylation (top) and corresponding Ponceau S staining (bottom) for WT strains of the indicated *ade2-n* locations. Source data are available in *Figure 5—source data 1*.

The online version of this article includes the following source data and figure supplement(s) for figure 5:

Source data 1. This file contains all the source data for *Figure 5* and related figure supplements.

Figure supplement 1. Interchromosomal contact frequency does not significantly impact recombination efficiency.

Figure supplement 2. Rad51 over-expression partially rescues the interchromosomal recombination defect of long-range resection deficient cells.

## Rad51 over-expression suppresses the interchromosomal HR defect of *exo1Δ sgs1Δ* cells

Rad51 is required for increased mobility of a damaged locus even though the checkpoint is active in Rad51-deficient cells (*Dion et al., 2012*; *Kalocsay et al., 2009*; *Miné-Hattab and Rothstein, 2012*; *Smith et al., 2018*). It has been suggested that Rad51 binding to resected ends stiffens them, thereby enhancing the homology search (*Miné-Hattab et al., 2017*). Since resection tracts are ~300 nt long in *exo1Δ sgs1Δ* cells, nucleation of Rad51 might be slower, resulting in less efficient interchromosomal HR. Expression of Rad51 from a high copy number plasmid partially suppressed the interchromosomal recombination defect of the *exo1Δ sgs1Δ* mutant (*Figure 5—figure supplement 2*). These data suggest that Rad51 is not completely saturated on the short overhangs, contributing to lack of mobility of the damaged site and reduced recombination efficiency.

## Discussion

Current models of HR include MRX-Sae2 catalyzed resection initiation, followed by long-range resection by Exo1 and/or Dna2-Sgs1. Here, we show that long-range resection is dispensable for DSB-induced HR when a repair template is located in close proximity on the same chromosome, suggesting that sufficient ssDNA is generated by MRX-Sae2 for Rad51-catalyzed repair. Long-range resection becomes crucial for efficient recombination when the donor allele is located on a different chromosome or greater than ~50–100 kb from the DSB site on the same chromosome where the DNA damage checkpoint is activated (*Figure 1*, *Figure 5*). The necessity for long-range resection in these scenarios is consistent with previous work showing a requirement for Exo1 and Sgs1-Dna2 in interchromosomal repair (*Gobbini et al., 2020*; *Guo et al., 2017*). Our results provide a possible explanation for why others have observed long-range resection-independent recombination (*Westmoreland and Resnick, 2016*; *Zakharyevich et al., 2010*). First, when G2 cells are treated with IR, a broken chromatid is likely in close proximity to its sister, facilitating the homology search and subsequent repair. Second, multiple DSBs are induced by IR and during meiosis, and even the limited resection at each of these sites is likely to cumulatively yield sufficient ssDNA to activate the checkpoint (*Gobbini et al., 2020*).

One caveat to our findings and those of others reporting long-range resection independent recombination is that resection tracts produced by MRX under physiological conditions are likely to be shorter than those formed in *exo1Δ sgs1Δ* cells due to engagement of the long-range resection machinery (*Mimitou and Symington, 2008*; *Zhu et al., 2008*). Thus, long-range resection may be playing a more important role than is apparent in cells lacking Exo1 and Sgs1.

Given that the DNA damage checkpoint is activated in scenarios where repair is delayed, long-range resection may act as a timing mechanism for checkpoint activation. Relatively quick repair circumvents the need for the checkpoint. However, if repair takes longer, long-range resection serves to create sufficiently long ssDNA tracts for checkpoint activation. In the absence of long-range resection, and therefore robust checkpoint activation, cells may proceed through to the next cell cycle with a broken chromosome, leading to cell death (*Figure 6*). Artificial G2/M arrest of *exo1Δ sgs1Δ* cells is insufficient to rescue the HR defect, indicating that the DNA damage checkpoint is responsible for eliciting effects other than cell cycle delay in order to promote HR. A more active role for the DNA damage checkpoint is supported by our finding that artificial induction of the checkpoint can partially rescue the interchromosomal repair defect of *exo1Δ sgs1Δ* cells (*Figure 4B*). The rescue we see is likely incomplete due to the short galactose induction time used to avoid a dominant negative effect of the checkpoint system. Additionally, we do not know whether inducing the checkpoint elsewhere in

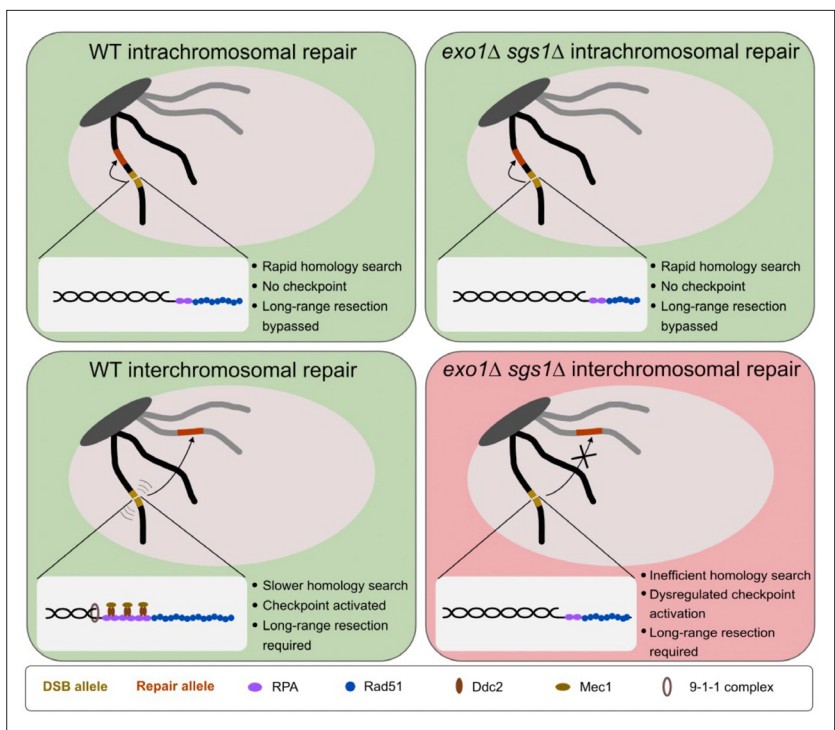

**Figure 6.** Model for the requirement for long-range resection in recombination. In intrachromosomal recombination, short resection tracts are sufficient to facilitate the homology search and repair due to the close proximity of the DSB allele (yellow) and repair allele (orange) (top panels). An intramolecular repair event is represented, but an inter-sister repair event is also possible. In interchromosomal recombination (or distal intrachromosomal recombination), the homology search takes longer. Therefore, resection tracts continue to be extended and the DNA damage checkpoint is activated in WT cells (bottom left panel). This would pause the cell cycle and activate mobility, facilitating the homology search and repair. However, in the absence of long-range resection, checkpoint activation is impaired, likely leading to a mobility defect and a failure to delay cell cycle progression (bottom right panel). This would result in cell death as cells would divide with a broken chromosome. The large light gray oval represents the nucleus and the dark gray oval represents clustered centromeres. Only two chromosomes are shown for simplicity.

the genome will have the same effect on mobility at the DSB site as the native checkpoint response. Finally, there is likely some effect of having short resection tracts that limits DSB mobility as well, possibly through inefficient Rad51 loading (*Figure 3D*, *Figure 5—figure supplement 2*).

The more substantial recombination deficiencies of *mec1Δ sml1Δ, rad53Δ sml1Δ, rad9Δ*, and *rad24Δ* cells in the interchromosomal versus the intrachromosomal context further supports the role of the checkpoint in the former, but not the latter scenario. As noted above, the failure of these mutants to facilitate efficient interchromosomal recombination is due to their checkpoint defect and not to a resection defect. The observation that loss of Rad9 or Rad24 in long-range defective cells is either epistatic or suppressive, respectively, supports the checkpoint defect in *exo1Δ sgs1Δ* cells. Restoring the checkpoint by expression of a Ddc2-Rad53 fusion protein in *rad9Δ* cells rescued the interchromosomal defect indicating that Rad9 is not required for HR per se and its function is to activate Rad53. It is also interesting to consider that the checkpoint, and therefore Mec1 and Rad9, are necessary for DNA damage-induced chromosome mobility (*Dion et al., 2012*; *Seeber et al., 2013*). Rad24 is linked to chromosome mobility since the loading of the 9-1-1 complex at resected DNA ends is necessary for Mec1 activation, and therefore checkpoint signaling (reviewed in: *Finn et al., 2012*). Given that interchromosomal recombination is likely to be more spatially challenging than proximal intrachromosomal repair, it is reasonable to think that the need for chromosome mobility would be increased for interchromosomal recombination. Consistent with this idea, we do observe a partial rescue of interchromosomal recombination in the absence of Exo1 and Sgs1 by overexpressing Rad51. This result likely reflects the fact that Rad51 is necessary for chromosome mobility and that this mobility helps facilitate repair (*Dion et al., 2012*; *Kalocsay et al., 2009*; *Miné-Hattab et al.,*

*2017*; *Miné-Hattab and Rothstein, 2012*; *Smith et al., 2018*). Rad51 loading stiffens and extends ssDNA (*Ogawa et al., 1993*; *Sung and Robberson, 1995*), possibly reducing the complexity of the homology search as has been shown recently for RecA in bacteria (*Wiktor et al., 2021*). The rate limiting step of Rad51 filament formation is the nucleation step, which requires ~6 Rad51 monomers, and increased concentrations of Rad51 promote nucleation in vitro (*Candelli et al., 2014*; *Miné et al., 2007*; *Paoletti et al., 2020*; *Qiu et al., 2013*). The increased Rad51 concentration likely increases the probability of nucleation on short resection tracts, thereby promoting filament polymerization for homology search. Interestingly, it was recently shown that distal global mobility is facilitated by and enhances HR and is dependent on Rad51 and Rad9, while proximal global mobility occurs independent of these factors and is dispensable for HR (*García Fernández et al., 2022*). Given that long-range resection is an important upstream step of both checkpoint activation and Rad51 loading, this result may explain why we observe that long-range resection is required for distal (interchromosomal) recombination but is dispensable for proximal (intrachromosomal) recombination.

We find that WT cells repair the I-*Sce*I break at high efficiency, independent of the donor location (*Figure 5A*, *Figure 5—figure supplement 1*). This is in contrast to findings that showed differences in both intra- and interchromosomal repair efficiency that correlated with contact frequency in WT cells (*Agmon et al., 2013*; *Lee et al., 2016*; *Wang et al., 2017*). Although the difference between these results and our own is unclear, it may be due to donor homology and/or use of HO versus I-*Sce*I. Donor size positively impacts repair efficiency (*Lee et al., 2016*), and our system contains significantly more homology than prior studies (3.7 kb vs 2 kb or 1.2 kb) (*Agmon et al., 2013*; *Lee et al., 2016*). The difference in cutting efficiency of the endonuclease may also influence repair proficiency as determined by plating assays. Since I-*Sce*I cleavage is less efficient than HO, some cells might divide on galactose-containing medium before the DSB is induced and survival of only one daughter cell would be required for colony formation leading to an over-estimation of repair efficiency. Based on our results, it seems that regardless of the donor position, sufficient end resection and checkpoint activation can facilitate repair, accounting for the fact that we only see differences in repair efficiency in the absence of long-range resection.

The failure to robustly rescue interchromosomal recombination deficiency of long-range resection deficient cells by using higher contact donors was unexpected at first, given previous work (*Lee et al., 2016*). However, even the higher contact frequency interchromosomal donors that were used are still relatively low compared to intrachromosomal contacts within Chr XV (*Lazar-Stefanita et al., 2017*). Hence, it might not be as surprising that we only observed improvement of recombination efficiency for the closest donor sites on the same chromosome. We also do not see an effect of centromere proximity on repair that was reported previously (*Wang et al., 2017*). However, this could be due to not using donors within 240 kb of the centromere. Nevertheless, our intrachromosomal data suggest that linear distance between the DSB site and donor site is the primary determinant of repair efficiency in the absence of long-range resection. Together, these results demonstrate that cells deficient for long-range resection are only proficient for recombination when the recombining sequences are in close proximity, but that WT cells can facilitate recombination independent of donor proximity.

Based on our results, it becomes clear that the importance of long-range resection does not necessarily lie in its ability to expose multiple kilobases of homology, although there may be certain contexts in which this is important. For example, long-range resection promotes the usage of more extensive stretches of homology, even if they are not in the immediate vicinity of the DSB. This may be especially important if a DSB occurs in a repetitive element, as long-range resection should suppress usage of short homologies between repeats and favor recombination with more extensive homologies in the surrounding sequence. Long-range resection may also promote a more efficient homology search by permitting invasion of multiple substrates (*Wright and Heyer, 2014*). However, this multi-invasion process has also been shown to be mutagenic, so there may be a trade-off (*Piazza et al., 2017*; *Ruiz et al., 2009*). Additionally, long-range resection has been shown to suppress telomere addition at slowly-repaired DSBs (*Chung et al., 2010*; *Lydeard et al., 2010*). Although we cannot exclude the possibility that loss of recombinants in the *exo1Δ sgs1Δ* interchromosomal assay is due to de novo telomere addition, it is unclear why this would not also occur in the intrachromosomal system.

We suggest that an additional function of long-range resection is to activate the checkpoint when repair is delayed, thereby increasing chromosome mobility and promoting the search for homology. This function may be relevant in a context in which recombination with the sister chromatid is inefficient,

thus necessitating repair with a non-allelic donor, potentially resulting in a compromise between cell survival and genome integrity. Despite the circumvention of long-range resection in scenarios of rapid repair, this process clearly serves an important purpose in multiple contexts and is supported by the evolutionary maintenance of two redundant pathways to create longer tracts of ssDNA.

# Materials and methods

**Key resources table**

| Reagent type (species) or resource | Designation | Source or reference | Identifiers | Additional information |
|---|---|---|---|---|
| strain, strain background (*Saccharoymces cerevisiae*, W303) | Various | Various | Various | See Materials and Methods section |
| antibody | α-Rad53 (Mouse, monoclonal) | M.Foiani | Clone EL7 | (1:500) |
| antibody | α-HA [12CA5] (Mouse, monoclonal) | Roche (thru Millipore Sigma) | SKU# 11583816001 | (1:1000) |
| other | α-mouse IgG kappa BP-HRP | Santa Cruz Biotechnology | Cat# sc-516102 | HRP-conjugated recombinant binding protein (1:5000) |
| recombinant DNA reagent | pRG205MX (DNA plasmid) | *Gnügge et al., 2016* | Available on Addgene (Plasmid #64535) | *LEU2* shuttle vector |
| recombinant DNA reagent | pAG25 (DNA plasmid) | Addgene | Cat# 35121 | *NatMX* plasmid |
| recombinant DNA reagent | *ade2-n-LEU2MX* (DNA plasmid) | This study | pLS515 | See Materials and methods section |
| recombinant DNA reagent | *ade2-n-NatMX* (DNA plasmid) | This study | pLS617 | See Materials and methods section |
| recombinant DNA reagent | pRS423 (DNA plasmid) | *Christianson et al., 1992* | Empty vector (EV) | |
| recombinant DNA reagent | pRS423-*RAD51* (DNA plasmid) | This study | pLS506 | See Materials and methods section |
| recombinant DNA reagent | pAFS52 (DNA plasmid) | *Straight et al., 1996* | | 256 x LacO plasmid with TRP1 marker |
| sequence-based reagent | Primers for ADE2 recombination assay and qPCR primers for measuring cutting efficiency and resection | Various | Various | See Materials and methods section |
| commercial assay or kit | MasterPure Yeast DNA Purification Kit | BiosearchTechnologies | Cat# MPY80200 | |
| commercial assay or kit | SsoAdvanced Universal SYBR Green Supermix | Bio-Rad | Cat# 1725274 | |
| commercial assay or kit | Qubit 1 X dsDNA High Sensitivity Assay Kit | Invitrogen | Cat# Q33231 | |
| commercial assay or kit | SuperSignal West Femto Max Sensitivity ECL | ThermoFisher | Cat# 34096 | |
| commercial assay or kit | Phusion High Fidelity DNA Polymerase kit | New England BioLabs | Cat# 0530 L | |
| chemical compound, drug | Trichloroacetic acid (TCA) | Sigma | Cat# T0699-100mL | |
| chemical compound, drug | Dimethyl Sulfoxide (DMSO) | Fisher Scientific | Cat# D128-1 | |
| chemical compound, drug | Nocodazole | AbMole | Cat# M3194 | |
| chemical compound, drug | 2% Bis Solution | Bio-Rad | Cat# 1610142 | |
| chemical compound, drug | 40% Acrylamide Solution | Bio-Rad | Cat# 1610140 | |
| chemical compound, drug | N,N,N',N'-Tetramethyl-ethylenediamine (TEMED) | Sigma | Cat# T9281-25mL | |

*Continued on next page*

*Continued*

| Reagent type (species) or resource | Designation | Source or reference | Identifiers | Additional information |
|---|---|---|---|---|
| chemical compound, drug | Ponceau S | Sigma | P3504-10G | |
| chemical compound, drug | β-estradiol | Sigma | Cat# E8875 | |
| software, algorithm | Prism V9.0 | GraphPad | | |
| other | Glass beads, acid washed | Sigma | Cat# G8772 | Beads for cell lysis (See Materials and Methods section) |
| other | AatII | New England BioLabs | Cat# R0117L | Restriction enzyme |
| other | BamHI-HF | New England BioLabs | Cat# R3136S | Restriction enzyme |
| other | BglII | New England BioLabs | Cat# R0144S | Restriction enzyme |
| other | EcoRV-HF | New England BioLabs | Cat# R3195S | Restriction enzyme |
| other | rCutSmart buffer | New England BioLabs | Cat# B6004S | Restriction enzyme buffer |
| other | NEBuffer r3.1 | New England BioLabs | Cat# B6003S | Restriction enzyme buffer |
| other | 1 kb ladder | New England BioLabs | Cat# N3232L | DNA size ladder |
| other | iBlot 2 PVDF Mini Stacks | Invitrogen | Cat# IB24002 | PVDF western membrane and dry transfer stack |
| other | FastPrep-24 5 G homogenizer | MP-Biomedicals | Cat# 6005500 | Sample prep system (See Materials and methods section) |

## Media and yeast strains

Complete yeast media contained 1% yeast extract, 2% peptone, 10 µg/mL adenine, and either 2% glucose (YPAD), or 2% raffinose (YPAR) as a carbon source. Galactose was added to YPAR to 2% final from a 20% stock for conditions of DSB induction. Synthetic media contained 1 X yeast nitrogen base, 1 X amino acid dropout mix, and 2% glucose or 2% raffinose.

All yeast strains are in the W303 background and are listed in *Supplementary file 1*: Yeast strains. Strains were constructed by standard genetic methods. Lithium acetate transformations were used to introduce deletion cassettes containing a marker of choice and homology arms flanking the gene to be deleted. Other strains were made by crossing, followed by tetrad dissection and marker selection.

For introducing the Chr III *ade2-n* cassette, a 3.7 kb fragment containing *ade2-n* and flanking genomic sequence was isolated from pAL78 (*Rattray and Symington, 1994*) by BamHI digest and cloned into a BamHI-digested pRG205MX to make pLS515. Then the plasmid was integrated into the *leu2* locus of an *ade2Δ* strain (LSY2584). The resulting strain was then crossed to LSY1738-3B, which contained the *ade2-I* allele at the native locus on Chr XV and $P_{GAL}$-*I-SceI* at the *lys2* locus, generating LSY4540-7B. For other inter- and intrachromosomal *ade2-n* integrations, the 3.7 kb *ade2-n* fragment was isolated from pLS515 by BamHI digest and cloned into a BamHI-digested pAG25 (NatMX) to generate pLS617. Oligos were designed to amplify *ade2-n-NatMX* with homology arms corresponding to the site of integration. Homology arms were extended to 80 bp total by two rounds of PCR (see *Supplementary file 2*: Oligonucleotides). These fragments were integrated into LSY1738-3B by standard LiAc-TE transformation. Integrations were confirmed by PCR screen of Nat[+] colonies.

For introducing the Ddc1-LacI Ddc2-LacI 256 x LacO checkpoint co-localization system into the interchromosomal strains, LSY5325-448A (WT) and LSY5326-252D (*exo1Δ sgs1Δ*) were first derived from a genetic cross with CBY88 (*Bonilla et al., 2008*). Then pAFS52 (256xLacO-*TRP1*) (*Straight et al., 1996*) was digested with EcoRV, which cuts within *TRP1*, and transformed into LSY5325-448A and LSY5326-252D to integrate the LacO array into the *TRP1* locus. Trp[+] transformants were subsequently screened by Southern blot using a *TRP1* probe and BglII digest of genomic DNA to identify those with a full-length integrated LacO array.

All yeast strains and plasmids are available by e-mail request to the corresponding author.

## Recombination plating assays

Cells were grown on YPAD (interchromosomal) or SC-Trp (intrachromosomal) agar plates at 30 °C for ~3 days. Single red colonies were picked and resuspended in 2 mL of YPAR media and incubated while shaking for 4 hr. Cultures were then centrifuged at 2000 x $g$ for 5 min. After removing supernatant, cells were resuspended in 200 μL of H$_2$O. Cells were then diluted to the appropriate plating dilution to ideally yield 100–200 colonies per plate (1:10,000 for most strains on YPAD, variable dilutions on YPA-GAL). A total of 200 μL of each dilution was plated onto 2 YPAD and 2 YPA-GAL plates. Plates were then incubated at 30 °C for 3–5 days. Colonies were counted and colony color (red/white) was noted. For the intrachromosomal strains with the *TRP1* marker between the repeats, YPA-GAL plates were replica plated to SC -Trp plates, which were grown for 2 days and counted to determine whether recombinants were Trp$^+$ or Trp$^-$.

To calculate survival frequency, an average of the colony count on the two YPA-GAL plates was divided by the average of the colony count on the two YPAD plates. Survival frequencies were determined for at least three independent cultures of each strain. The average survival frequency for each genotype is plotted in the bar graphs. Significance was determined by a two-tailed t-test. To determine the relationship between survival frequency and contact frequency or distance in the intrachromosomal assays, a two-tailed Spearman correlation analysis was applied in Prism using a 95% confidence interval. Additionally, a nonlinear, one-phase decay model was applied to each data set and is plotted in *Figure 5B*.

For Rad51 overexpression experiments, WT (LSY4540-7B) and *exo1Δ sgs1Δ* (LSY4614-2-2D) interchromosomal assay strains were transformed with pLS506 (pRS423-*RAD51*) or pRS423 (EV). Plating assays were carried out as described above, except that strains were grown in media lacking histidine to maintain the plasmid. Survival frequency was determined based on colony number on SC -His+GAL as compared to SC -His plates.

## Repair kinetics

A single red (Ade$^-$) colony was picked from a YPAD (interchromosomal) or SC -Trp (4 kb intrachromosomal) plate into 5 mL of SC -Trp (w/ 2% glucose and 10 μg/mL adenine) (intra) or the same media supplemented with 80 μg/mL Trp (inter). Cultures were grown for 8 hr and cell concentrations were determined. Cultures were diluted to 7x10$^4$ cells/mL in 50 mL of SC -Trp (w/ 2% raffinose and 10 μg/mL adenine) (intra) or 50 mL of the same media supplemented with 80 μg/mL Trp (inter). Raffinose cultures were grown overnight. A sample was taken prior to DSB induction (t$_0$), galactose was added to 2% final concentration to induce I-*Sce*I, and samples were collected at the indicated times after induction. ~3.5 × 10$^7$ cells were taken at each timepoint.

For G2-arrested conditions, cells were grown as described above and exponentially growing raffinose cultures were arrested with nocodazole at a final concentration of 20 μg/mL, plus DMSO at 1% final. For this purpose, DMSO was added to 1% final and nocodazole was added to 13.3 μg/mL final and cultures were grown for 1 hr. An additional 6.7 μg/mL of nocodazole was added, and cultures were grown for another hour. Arrest was confirmed by checking for large-budded cells under the microscope. DSB induction and timepoint collection was performed as described above.

For checkpoint co-localization experiments, cells were grown as described above and exponentially growing raffinose cultures were induced for DSB and checkpoint co-localization simultaneously by addition of galactose to 2% final. After the indicated time, glucose was added to 2% final to repress expression of I-*Sce*I and the checkpoint.

Genomic DNA was extracted with the MasterPure Yeast DNA Purification Kit (Biosearch Technologies) and DNA concentrations were measured using the Qubit Flex fluorometer and 1 x HS dsDNA assay kit (Invitrogen). PCR amplification followed by restriction digestion was used to measure repair kinetics. Twenty μL Phusion (NEB) reactions were set up according to manufacturer's protocol with primers MK193, MK197, MK238, and MK239 (*Supplementary file 2*), 100 ng of genomic DNA, and 30 cycles. 1 μL of PCR product was then used in a 20 μL digest reaction with AatII restriction enzyme and rCutSmart buffer (NEB). Digestion products were separated on 0.8% agarose gels made with 0.5 X TBE. Percent repair is a ratio of band intensity of repair products normalized to control product in the same reaction. The band intensity at t$_0$ was subtracted from all timepoints and the resulting intensities were used to calculate percent repair, setting the Ade$^+$ products at 100% repair.

## Rad53 Western Blotting

Culturing was performed as described for the repair kinetics experiments. ~1.4 × 10$^8$ cells were taken at each timepoint. Protein was extracted using Trichloroacetic acid (TCA) precipitation. Cell pellets were washed once in 500 µL of 20% TCA and then resuspended in 200 µL of 20% TCA and transferred to 2 mL screw cap tubes on ice. An equal volume of acid-washed glass beads (Sigma) was added. Cells were physically lysed in the FastPrep-24 5 G homogenizer (MP-Biomedicals) at 4 °C. The machine was run at 10 m/s for 20 s for three rounds. Supernatants were transferred to a new Eppendorf tube, beads were washed twice with 200 µL of 5% TCA, and all supernatants were combined and centrifuged at 3000 rpm for 10 min. The supernatant was discarded and cell pellets were resuspended in 1 x Laemmli SDS-PAGE loading buffer (50 mM Tris-HCl pH6.8, 2% SDS, 10% glycerol, 0.02% Bromophenol blue, 5% 2-mercaptoethanol). Samples were then boiled for 5 min before loading on a 10% Acrylamide/ Bis-acrylamide gel. Protein bands were transferred to PVDF membranes using the iBlot 2 Transfer Stacks, mini and the iBlot 2 Gel Transfer Device. Membranes were then stained with Ponceau S for 10 min, followed by two to three 5 min rinses in water until protein bands were apparent. Stained membranes were imaged and then further de-stained in 1 x TBS-T (TBS +0.1% Tween) to remove the remaining dye. Membranes were gently shaken in blocking solution (5% milk in TBS-T) for 1 hr at room temperature on an orbital shaker. Then the blocking solution was removed and the primary antibody was added (Anti-Rad53 EL7 antibody (gift from M. Foiani) diluted 1:500 in fresh blocking solution). For detection of Rad53-HA, anti-HA antibody (12CA5 from mouse (Roche)) diluted 1:1,000 in blocking solution was used. Membranes were incubated in the primary antibody overnight at 4 °C while gently rocking on a nutator. The following day, the primary antibody was removed, membranes were washed in TBS-T 3 x for 5 min each. The secondary antibody was then added (anti-mouse IgG kappa binding protein [Santa Cruz Biotechnology] diluted 1:5000 in blocking solution) and incubated for 2 hr at room temperature while rocking on a nutator. The secondary antibody was removed and membranes were washed in TBS-T 3 x for 5 min each. One final 5-min wash with TBS was performed to remove Tween. Membranes were developed using a 1:1 ratio of SuperSignal West Femto Max Sensitivity ECL reagents (Thermo Fisher) for 2 min. Membranes were then exposed to film and developed in a Kodak X-OMAT processor.

## Quantitative PCR-based resection assay

Resection assays were carried out as previously described (*Gnügge et al., 2018*; *Zierhut and Diffley, 2008*). The LexO-HO strains were used in G2-arrested conditions and DSB formation was induced by addition of 2 µM β-estradiol (diluted from a 10 mM β-estradiol in ethanol stock) (*Gnügge and Symington, 2020*).

## Calculating contact frequencies

Contact frequencies were calculated by using a ±20 kb window around the DSB site and a ±30 kb window around the repair locus as in *Lee et al., 2016*. Data sets from *Duan et al., 2010* (HindIII data) and *Lazar-Stefanita et al., 2017* (GSM2417285_asynchronous_I.filt.5000.rebin.csv) were analyzed. Contact frequencies extracted from *Lazar-Stefanita et al., 2017* were used in *Figure 5B*. Percentile ranking of contact frequencies was determined by making a ±30 kb sliding window across each chromosome. Then, contact frequencies for each of these windows were ranked and broken down into percentiles.

# Acknowledgements

We thank R Rothstein, D Toczyski and M Foiani for gifts of yeast strains, plasmids and anti-Rad53 antibodies. We thank J Haber, W Holloman and members of the Symington lab for review of the manuscript and helpful discussions. This work was supported by grants from the National Institutes of Health (R35 GM126997, T32 GM008798 and T32 CA265828).

# Additional information

### Funding

| Funder | Grant reference number | Author |
|---|---|---|
| National Institute of General Medical Sciences | R35 GM126997 | Lorraine S Symington |
| National Institute of General Medical Sciences | T32 GM008798 | Michael T Kimble |
| NIH/NCI | T32 CA265828 | Matthew J Johnson |

The funders had no role in study design, data collection and interpretation, or the decision to submit the work for publication.

### Author contributions

Michael T Kimble, Conceptualization, Data curation, Formal analysis, Validation, Investigation, Methodology, Writing – original draft, Writing – review and editing; Matthew J Johnson, Conceptualization, Data curation, Formal analysis, Investigation, Methodology, Writing – review and editing; Mattie R Nester, Investigation, Methodology, Writing – review and editing; Lorraine S Symington, Conceptualization, Formal analysis, Supervision, Funding acquisition, Investigation, Methodology, Writing – original draft, Project administration, Writing – review and editing

### Author ORCIDs

Michael T Kimble ⓘ http://orcid.org/0000-0001-8042-2868
Matthew J Johnson ⓘ http://orcid.org/0000-0002-4312-5618
Lorraine S Symington ⓘ http://orcid.org/0000-0002-1519-4800

### Decision letter and Author response

Decision letter https://doi.org/10.7554/eLife.84322.sa1
Author response https://doi.org/10.7554/eLife.84322.sa2

# Additional files

### Supplementary files

• Supplementary file 1. Yeast Strains. All yeast strains used in this study.

• Supplementary file 2. Oligonucleotides. Oligonucleotides used for recombination assays and generation of ectopic recombination strains.

• MDAR checklist

### Data availability

All data generated or analyzed during this study are included in the manuscript and supporting files; Source data files have been provided for Figures 1-5 and associated figure supplements.

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
