## [Editor Report]

This study shows that long-range resection is important for recombination between distal, but not proximal, homologous sequences. It is thus proposed that a major role of long resection of a double strand break mediated by Sgs1 and Exo1 is to activate the DNA damage checkpoint to allow the chromosomal mobility needed for the DNA ends to find a distant homologous sequence for repair via homologous recombination. Consistently, *exo1 sgs1* mutants show defects specifically in interchromosomal homologous recombination. The study adds a new biological meaning to the role of long DNA resection, providing a new mechanism for the control of homologous recombination.

---

## [Decision Letter]

**Decision letter after peer review:**

Thank you for submitting your article "Long-range DNA end resection supports homologous recombination by checkpoint activation rather than extensive homology generation" for consideration by *eLife*. Your article has been reviewed by 3 peer reviewers, one of whom is a member of our Board of Reviewing Editors, and the evaluation has been overseen by Kevin Struhl as the Senior Editor. Reviewers opted not to reveal their identity.

Essential revisions:

1. Related to the rescue of interchromosomal recombination using the Ddc2-Rad53 chimera, it would be important to test the rescue of rad24Δ exo1Δ sgs1Δ mutants with the chimera. In Figure 4A, the authors show that Ddc2-Rad53 fusion can rescue the defects in rad9Δ, but this construct fails to rescue the defects in exo1Δ sgs1Δ (Figure S4A). The authors propose that the lack of rescue is related to the short tracks in sgs1Δ exo1Δ cells. If this is the case, Ddc2-Rad53 expression should robustly, although partially, rescue recombination defects in rad24Δ exo1Δ sgs1Δ cells given the longer ssDNA tract that should be able to sustain Ddc2-Rad53 recruitment and activation. The authors may want to consider fusing a Mec1-activating domain (MAD) from Ddc1 into their Ddc2-Rad53 chimera to make sure Mec1 is robustly activated in the absence of 9-1-1 loading.

2. Given the difficulties to demonstrate a robust rescue of interchromosomal recombination in exo1Δ sgs1Δ cells with strategies for forced Rad53 recruitment, the authors could also consider alternative strategies. For example, it was previously shown that mutation of serine 486 to alanine (S486A) in Slx4 leads to persistent Rad9 recruitment, and persistent Rad53 hyperactivation (Ohouo et al. 2013; Cussiol et al., 2015; Dibitetto et al., 2016). Therefore, this mutation is expected to robustly rescue interchromosomal recombination defects in exo1Δ sgs1Δ cells. The experiment to test this should be feasible (S486A mutation is also not synthetic lethal with deletion of SGS1 (unpublished data)) and straightforward.

3. The double band pattern of Rad53 in the exo1Δ sgs1Δ mutant should be investigated further since the constitutive Rad53 activation status could be relevant for the recombination outcomes. Is the shift dependent on Rad9? The authors should discuss more what would be the implications of a constitutively active Rad53 in their system. Moreover, because the authors show constant Rad53 phosphorylation in exo1Δ sgs1Δ cells, it would be useful to monitor Rad53 phosphorylation-induced mobility shift in rad24Δ exo1Δ sgs1Δ and check if the rescue is, or not, associated with altered checkpoint signaling.

4. Related to the previous point, in Figure 4C, left panel, exo1Δ sgs1Δ shows Rad53 phosphorylation at 0h timepoint, consistent with Figure 2D. However, the phosphorylation disappears at 0h timepoint when the co-localization system is functional. Since the authors mention the dominant-negative effect, is it possible that the constant Rad53 phosphorylation in exo1Δ sgs1Δ is toxic to distal recombination, so when the dominant-negative effect dampens Rad53 phosphorylation recombination gets partially rescued? (it has been demonstrated that rad53-K227A can suppress the sensitivity of exo1Δ sgs1Δ cells to DNA damage (Gobbini et al., 2020)). In addition, the authors show that Ddc2-Rad53 expression enhances the recombination defects in exo1Δ sgs1Δ (Figure S4A), consistent with the idea that constant Rad53 activation in exo1Δ sgs1Δ may impair the repair process. Discuss.

---

## [Author Response]

Essential revisions:1. Related to the rescue of interchromosomal recombination using the Ddc2-Rad53 chimera, it would be important to test the rescue of rad24Δ exo1Δ sgs1Δ mutants with the chimera. In Figure 4A, the authors show that Ddc2-Rad53 fusion can rescue the defects in rad9Δ, but this construct fails to rescue the defects in exo1Δ sgs1Δ (Figure S4A). The authors propose that the lack of rescue is related to the short tracks in sgs1Δ exo1Δ cells. If this is the case, Ddc2-Rad53 expression should robustly, although partially, rescue recombination defects in rad24Δ exo1Δ sgs1Δ cells given the longer ssDNA tract that should be able to sustain Ddc2-Rad53 recruitment and activation. The authors may want to consider fusing a Mec1-activating domain (MAD) from Ddc1 into their Ddc2-Rad53 chimera to make sure Mec1 is robustly activated in the absence of 9-1-1 loading.

As suggested by the reviewers, we tested rescue of *rad24Δ* derivatives by the Ddc2-Rad53 fusion protein. As expected, Ddc2-Rad53 was unable to suppress the *rad24Δ* HR defect because it is unable to rescue the checkpoint defect caused by loss of Rad24 (Lee et al., 2004). However, we did observe a partial rescue of the severe recombination defect observed in *exo1Δ sgs1Δ* Ddc2-Rad53 cells, these data are now included in Figure 4—figure supplement 1.

2. Given the difficulties to demonstrate a robust rescue of interchromosomal recombination in exo1Δ sgs1Δ cells with strategies for forced Rad53 recruitment, the authors could also consider alternative strategies. For example, it was previously shown that mutation of serine 486 to alanine (S486A) in Slx4 leads to persistent Rad9 recruitment, and persistent Rad53 hyperactivation (Ohouo et al. 2013; Cussiol et al., 2015; Dibitetto et al., 2016). Therefore, this mutation is expected to robustly rescue interchromosomal recombination defects in exo1Δ sgs1Δ cells. The experiment to test this should be feasible (S486A mutation is also not synthetic lethal with deletion of SGS1 (unpublished data)) and straightforward.

As suggested by the reviewers, we incorporated *SLX4-FLAG::*K*anMX* and *slx4-S486AFLAG::KanMX* cassettes into the wild-type interchromosomal recombination reporter strain. However, we discovered that the strains exhibited a growth defect on YP-Gal plates, with <1% viability relative to growth on YPD. The colonies grown on YPD exhibited white sectoring from leaky expression of I-SceI indicating normal levels of recombination, suggesting that the growth defect on YP-Gal was not due to HR deficiency. To obviate the need for galactose induction, we turned to a β-estradiol-inducible Cas9 construct (Al-Zain et al., 2023) with a gRNA targeted to the I-SceI cut site that we developed for another project. The strains with *KanMX* marked *SLX4* and *slx4-S486A* alleles were crossed to an *exo1Δ sgs1Δ* derivative with the alternate endonuclease induction system. Although the *SLX4* and *slx4-S486A* strains exhibited similar recombination frequencies to WT, we failed to detect a suppression of the *exo1Δ sgs1Δ* recombination defect by *slx4-S486A* (Author response image 1). Consistently, *slx4-S486A* did not rescue the Rad53 phosphorylation defect of the *exo1Δ sgs1Δ* mutant. We suspect that the defect in Mec1 signaling caused by short resection tracts results in reduced Rad9 accumulation in the *exo1Δ sgs1Δ* mutant. Surprisingly, we did not observe hyperactivation of Rad53 by the *slx4-S486A* mutation in the WT W303 strain background. We do not have an explanation for this discrepancy other than a potential strain background difference, and for this reason did not include these data in the revised manuscript.

**Author response image 1. sa2fig1:** Interchromosomal recombination in the presence of *slx4S486A*. (A) Results from the recombination plating assay using a modified interchromosomal strain expressing Cas9 and an I*Sce*I-targeted gRNA rather than GAL-I-*Sce*I. Bars represent average survival frequency and error bars indicate standard deviation of five biological replicates per genotype. ns, not significant. (B) Rad53 western blot (top) and Ponceau S staining (bottom) for the indicated strains 0 and 4 h ajer Cas9 induction.

3. The double band pattern of Rad53 in the exo1Δ sgs1Δ mutant should be investigated further since the constitutive Rad53 activation status could be relevant for the recombination outcomes. Is the shift dependent on Rad9? The authors should discuss more what would be the implications of a constitutively active Rad53 in their system. Moreover, because the authors show constant Rad53 phosphorylation in exo1Δ sgs1Δ cells, it would be useful to monitor Rad53 phosphorylation-induced mobility shift in rad24Δ exo1Δ sgs1Δ and check if the rescue is, or not, associated with altered checkpoint signaling.

As the reviewer has suggested, we measured Rad53 phosphorylation in both *exo1∆ sgs1∆ rad9∆* and *exo1∆ sgs1∆ rad24∆* cells (Figure 2—figure supplement 2). In both cases, the phospho-shij present in *exo1∆ sgs1∆* cells in the absence of damage was eliminated by additional deletion of *RAD9* or *RAD24.* Furthermore, there is no DNA-damage-dependent shij in *exo1∆ sgs1∆ rad24∆* cells. Given that *exo1∆ sgs1∆ rad9∆* and *exo1∆ sgs1∆ rad24∆* cells show a similar Rad53 phosphorylation pattern, but only *exo1∆ sgs1∆ rad24∆* shows a partial rescue of recombination, we favor the conclusion that this is due to the partial rescue of resection in *exo1∆ sgs1∆ rad24∆* cells as shown in Figure 3—figure supplement 1. Resection 640bp from the DSB in *exo1∆ sgs1∆ rad9∆* cells is the same as *exo1∆ sgs1∆* cells (Figure 3—figure supplement 1D).

4. Related to the previous point, in Figure 4C, left panel, exo1Δ sgs1Δ shows Rad53 phosphorylation at 0h timepoint, consistent with Figure 2D. However, the phosphorylation disappears at 0h timepoint when the co-localization system is functional. Since the authors mention the dominant-negative effect, is it possible that the constant Rad53 phosphorylation in exo1Δ sgs1Δ is toxic to distal recombination, so when the dominant-negative effect dampens Rad53 phosphorylation recombination gets partially rescued? (it has been demonstrated that rad53-K227A can suppress the sensitivity of exo1Δ sgs1Δ cells to DNA damage (Gobbini et al., 2020)). In addition, the authors show that Ddc2-Rad53 expression enhances the recombination defects in exo1Δ sgs1Δ (Figure S4A), consistent with the idea that constant Rad53 activation in exo1Δ sgs1Δ may impair the repair process. Discuss.

We believe that the reason for the lack of Rad53 phosphorylation at the 0 hr timepoint with the co-localization system is not due to the dominant negative effect, but rather due to the fact that Ddc1 is only expressed under galactose conditions (endogenous *DDC1* is deleted; *GAL*-*DDC1*LacI is only expressed in the presence of galactose). As we showed in response to the previous reviewer comment, the Rad53 phosphorylation observed in *exo1∆ sgs1∆* cells in the absence of damage is Rad24-dependent, and therefore Ddc1 dependent. We do see an induction of Rad53 phosphorylation ajer induction of the co-localization system with a concomitant rescue of recombination, supporting our claim that the recombination defect in *exo1∆ sgs1∆* cells is connected to its checkpoint deficiency.

*DDC2-RAD53* has replaced *RAD53* in our strains and is therefore under the control of the endogenous *RAD53* promoter. Therefore, *DDC2-RAD53* will be expressed the same as native *RAD53* and will not be constitutively activated. It will only bypass the need for Rad9-mediated recruitment of Rad53 to sites of damage. It is unclear why the fusion causes an increased recombination defect in *exo1∆ sgs1∆* cells, however we determined it is not due to further decreasing resection (Figure 4—figure supplement 1).